

# The effect of lossy compression of numerical weather prediction data on data analysis: a case study using enstools-compression 2023.11

Oriol Tintó Prims[1], Robert Redl[1], Marc Rautenhaus[2], Tobias Selz[1], Takumi Matsunobu[1], Kameswar Rao Modali[2], and George Craig[1]

[1]Meteorological Institute, Ludwig Maximilian University of Munich, Munich, 80333, Germany
[2]Regional Computing Centre, Visual Data Analysis Group, Universität Hamburg, Hamburg, 20146, Germany

**Correspondence:** Oriol Tintó Prims (oriol.tinto@lmu.de)

**Abstract.** The increasing amount of data in meteorological science requires effective data reduction methods. Our study demonstrates the use of advanced scientific lossy compression techniques to significantly reduce the size of these large datasets, achieving reductions ranging from 5x to over 150x, while ensuring data integrity is maintained. A key aspect of our work is the development of the 'enstools-compression' Python library. This user-friendly tool simplifies the application of lossy compression for Earth scientists and is integrated into the commonly used NetCDF file format workflows in atmospheric sciences. Based on the HDF5 compression filter architecture, 'enstools-compression' is easily used in Python scripts or via command line, enhancing its accessibility for the scientific community. A series of examples, drawn from current atmospheric science research, shows how lossy compression can efficiently manage large meteorological datasets while maintaining a balance between reducing data size and preserving scientific accuracy. This work addresses the challenge of making lossy compression more accessible, marking a significant step forward in efficient data handling in Earth sciences.

## 1 Introduction

In parallel with advances in observation instruments and computing resources, the speed at which new data is generated keeps increasing year after year. Science, geosciences, and weather are no exception (Zhao et al., 2020). While storage technology has also improved, the pace has been slower. Looking at how memory and CPU speeds evolved, there is a difference of orders of magnitude (Alted, 2010). With storage systems, the difference is even bigger. Yet, even though these IO systems did not become fast or big enough, they became cheap. That allowed users to overcome the storage capacity problem by buying more hardware. If at some point data generation outgrows disk cost reduction, a bigger proportion of the budget will have to go to storage. At this point, storage can threaten the feasibility of scientific projects. Although solving this problem requires much more than a single solution (Lawrence et al., 2019), one ingredient that can contribute to alleviating it is the adoption of data reduction techniques.

There are two ways to reduce the storage required for a dataset: reducing the number of values being stored or reducing the number of bits used to represent the same amount of values. When only specific variables are saved, and only at specific time steps, or when similar actions are taken, the first approach is being applied. Utilizing smaller data types, or employing other



methods to reduce the bit-per-value ratio, signifies the application of the second approach. While the first approach is common
in geosciences, the second one has been less used. Data compression, which is the subject of this paper, falls into this second
category.

The community has developed both lossless (Collet, 2020; Collet and Kucherawy, 2021; Deutsch and Gailly, 1996) and lossy
(Lindstrom, 2014; Liang et al., 2018; Zhao et al., 2020; Liu et al., 2021; Ballester-Ripoll and Pajarola, 2015; Ballester-Ripoll
et al., 2020; Klöwer et al., 2021; Düben et al., 2019) data compression methods. Lossless methods leverage redundancies to
reduce size. There is no loss of information and the original data is bit-to-bit recoverable. In contrast, with lossy methods, the
recovered data is not bit-to-bit identical anymore. Lossless methods may initially appear ideal, yet the potential compression
ratios that can be achieved for weather and Earth sciences data are very limited. In geosciences, the vast majority of disk
resources are committed to storing arrays of real numbers. Typically, these are stored using floating-point representations.
Despite the inherent uncertainty, a floating-point representation of a real number will consume a fixed number of bits. Hence,
when a quantity bears uncertainty, all bits are stored even though only a select few may carry significant value. Since most
geoscientific data has some uncertainty, some of the bits that end up stored are meaningless (Zender, 2016). These bits are not
only meaningless, but their randomness also makes them non-compressible. So, while the phrase "loss of information" may
initially sound intimidating, discarding useless data seems to be more of a solution than a problem.

In geosciences the widespread adoption of lossy methods has not happened yet, with one notable exception. The GRIB
format which is widely used in the weather community uses a lossy method. It applies a linear quantization to reduce the
number of bits per value. However, because for many variables the distribution of the values does not fit very well with a
linear distribution other methods would be better suited (Klöwer et al., 2021). The basic idea behind these methods is that
models and measurements generate false precision that results in meaningless data bits (Zender, 2016). Since these bits are
meaningless, getting rid of them should not result in a degradation of data quality. The difference between these methods is
how the non-significant bits are discarded. The most naive approach is bit-shaving (Caron, 2014), which consists of setting all
the bits after a certain position to 0. The same thing but setting all bits to 1 is known as bit setting. In the case of data arrays
truncating the values always in the same direction might affect statistical quantities. To solve that, bit grooming was proposed.
What it does is alternate between bit-shaving and bit-setting (Zender, 2016). On their own, these methods do not provide any
benefit in terms of data size, however, they allow to get higher compression ratios in a posterior lossless compression because
they help to get rid of meaningless uncompressible random bits (Zender, 2016). Besides these methods, other scientific lossy
compressors combine different methods to achieve higher compression ratios while allowing error control. By combining
different algorithms, these compressors achieve higher compression ratios for similar errors (Delaunay et al., 2019; Klöwer
et al., 2021).

Lossy compression is widely used in non-scientific applications. It is the standard for multimedia data. Usually, the algo-
rithms exploit human perception to maximize the compression ratio. For example, the image format Portable Network Graphics
has design choices that are based on how humans perceive images (Boutell, 1997). Also, most images are represented using
different channels with a limited bit depth. Such methods may not be satisfactory in the case of scientific data, where it is
important to have quantitative control of the errors.



To tackle the singularities of scientific data, a few active projects have attempted to develop suitable compressors. To the authors' knowledge, the list of the most interesting of these projects includes SZ (Liang et al., 2018; Zhao et al., 2020; Liu et al., 2021), ZFP (Lindstrom, 2014), FZIP (Lindstrom and Isenburg, 2006), THRESH (Ballester-Ripoll and Pajarola, 2015; Ballester-Ripoll et al., 2020). These compressors rely on different methods but pursue the same goal: high compression ratios with fine error control. Because the different projects have very similar objectives, there was an initiative to create a benchmark to facilitate comparison between them (Zhao et al., 2020). Along with this benchmark, few publications have compared these compressors (Zhao et al., 2020; Lindstrom, 2017). Since one of the objectives of these lossy compressors is to have control over the errors that are introduced, it is important to mention that the different methods used result in different error distributions (Lindstrom, 2017). Looking at the latest research on the topic, there are very promising works on the usage of auto-encoders for scientific data compression (Donayre Holtz, 2022; Liu et al., 2021). While preliminary results suggest that this approach can be very competitive for low bit-rates, these are still very slow compared to the alternatives and are not made public in a usable way.

Some work has been done to address the use of scientific lossy compressors with Earth sciences data (Klöwer et al., 2021; Poppick et al., 2020; Baker et al., 2014, 2016, 2017, 2019; Düben et al., 2019). A first conclusion is that variables with different distributions should be evaluated with different metrics. (Poppick et al., 2020; Baker et al., 2014). The different papers introduce different analysis methods: Klöwer et al. (2021) introduces the idea of using information theory to find the "real information content" while Baker et al. (2017) suggests that the errors introduced by lossy compression should not be statistically distinguishable from the natural variability of the climate system. In summary, the literature supports the idea that, when implemented correctly, lossy compression methods can safely help to reduce storage needs. To answer the question of what implemented correctly means, Baker et al. (2016) notes that the considerations that users need to make when applying lossy compression are not much different than other necessary choices like the grid resolution, data output frequency, among other factors, which will affect the results of our simulations. To illustrate how we have already been making trade-offs,consider the three-dimensional variables on pressure levels from the widely used ERA5 dataset. While the model simulation uses 137 vertical model levels, many users use the data reduced to 37 pressure levels. Similarly, the internal simulation time step is much shorter than the hourly outputs that are published. That is an example of a compromise between information and storage.

Along with achieving an effective data reduction, to facilitate widespread adoption it is imperative that productivity remains unaffected. Weather and Earth sciences communities heavily rely on the netCDF format (Rew et al., 1989) to store data. The newest version, netCDF-4, can use HDF5 (Koranne, 2010) as a back-end. One of the features of HDF5 is the possibility to use filters. By using filters, HDF5 can process the data on its way to/from the disk. In this way, one can make data-compression transparent to users, allowing them to keep the same file format and making explicit deflation not necessary. Moreover, developers of some state-of-the-art compressors have implemented their own HDF5 filter plugins. Delaunay et al. (2019) evaluated the usage of HDF5 filters with geoscientific data showing its feasibility. However, they evaluated for one set of compression parameters for all variables in the dataset, concluding that the digit-rounding is preferable. This leaves open the question of whether individual compression specifications for each variable can improve the results.



Tools to apply lossy compression to weather data already exist and are mature enough to be adopted. Our work pursues the goal of filling the knowledge gaps so users can start compressing their weather and other Earth sciences data. The objective of this publication is to advance toward that goal by enabling scientists to compress their existing datasets and create new ones that are directly compressed. Additionally, this work aims to ensure that scientists can seamlessly utilize the compressed data. Essentially, the intent is to facilitate a seamless integration of lossy compression into the research workflows of the weather and Earth science communities. From our perspective and looking at the literature, the missing gaps are (1) the difficulty to use existing lossy compressors and (2) the difficulty to decide which are appropriate compression specifications. To address these missing gaps, this article aims to achieve the following objectives: (1) providing a novel tool for straightforward use of lossy compression with NetCDF, (2) providing a method that computes optimal compression parameters based on user-specified quality metrics, and (3) evaluating our approach based on a number of use cases drawn from current research applications.

## 2 A user-friendly Python tool to use lossy compression with NetCDF files

We provide a user-friendly Python implementation that facilitates straightforward integration of lossy compression into workflows using the NetCDF file format, widely used in the atmospheric sciences. Our implementation is integrated into the "enstools-compression" Python package (Tintó-Prims, 2022a) and based on the HDF5 compression filter architecture (The HDF Group, 2023). It can be used from within Python scripts as well as from the command line. The source code is provided as open-source along with this article (Redl et al., 2022; Tintó-Prims, 2022a, b).

### 2.1 Exemplary compression schemes: ZFP and SZ

For this study, we selected the lossy compression algorithms ZFP (Lindstrom, 2014) and SZ (Liang et al., 2018; Zhao et al., 2020; Liu et al., 2021). Both offer competitive compression ratios with strict control of the errors, and for both HDF5 filters are available. Also, both algorithms were previously analyzed and compared by Zhao et al. (2020, providing information on compression ratio, speed, and further metrics for different scientific datasets) and Lindstrom (2017, providing information on error distributions of compressed floating-point data defined on structured grids). While outside the scope of this article, it is straightforward to extend our framework with further compression algorithms for which HDF5 compression filters are available.

ZFP (Lindstrom, 2014) is a transform-based compressor designed for 3-D data. Its compression algorithm consists of five steps, applied to fixed blocks of 4x4x4 grid points: (1) aligning the values of each block to a common exponent, (2) converting from a floating point to a fixed point representation, (3) decorrelating the values by doing a block transformation, (4) ordering the transform coefficients, and (5) applying an embedded coding algorithm. The ZFP algorithm offers four modes of lossy compression: (1) rate, (2) precision, (3) accuracy, and (4) an "expert mode". When using the rate mode, the bit rate of the resulting compressed file can be selected, i.e., the size of the compressed file can be specified. The precision mode preserves a user-specified number of bits of the original data, the file size is adapted accordingly and is not known in advance. The accuracy mode ensures that the errors introduced are smaller than a user-defined threshold. The expert mode allows the user to fine-tune



the ZFP algorithm with four parameters: minimum and maximum number of compressed bits per block, maximum precision in terms of encoded bit planes, and the smallest exponent value to control accuracy. Details are available in Lindstrom (2014).

SZ is a prediction-based compressor, and like ZFP designed for multi-dimensional data. Its compression algorithm consists of four steps: (1) value prediction using a Lorenzo predictor or a linear-regression-based predictor, (2) application of linear quantization, (3) variable-length encoding, and (4) lossless compression. The predictor is automatically selected based on the

compressed data. The algorithm is also applied block-wise, using blocks of 6x6x6 grid points for 3-D data and 12x12 blocks for 2-D data. The SZ algorithm provides three modes: (1) absolute, (2) relative, and (3) point-wise-relative. The absolute mode uses a user-defined absolute error threshold, and like ZFP's accuracy mode it ensures that errors are smaller than the provided threshold. For the relative mode a global relative error threshold is provided; the corresponding absolute error is computed with respect to the range of all data values in the dataset. The point-wise-relative mode uses a relative error with respect to each

data point's individual value. To illustrate the difference between relative and point-wise-relative modes, consider a dataset with values ranging from 1 to 1001. When using the relative mode with a global relative error of 1%, the compressed dataset will contain errors smaller than $(1001 - 1) \cdot 0.01 = 10$. When using the point-wise-relative mode with the same threshold, each data point in the compressed dataset will have an error smaller than 1% of its original value. Hence, values at the higher end of the range will have absolute errors smaller than 10, and those at the lower end will have absolute errors smaller than 0.01.

Note that in the example a global absolute error of 10 can lead to small positive values less than 10 becoming negative in the compressed representation. Hence, the point-wise-relative mode must be used when data values must remain strictly positive. Details are available in Liang et al. (2018).

In addition to lossy compression using ZFP and SZ, we integrate lossless compression for use cases where bit-to-bit reproducibility is required. For lossless compression, we use the BLOSC (Team) library that provides a number of state-of-the-art

algorithms. Lossless compression, however, is not further investigated in this work.

## 2.2 Compression Specification Format

The use of HDF5 compression filters requires the use of low-level programming interfaces, which demand a comprehensive understanding of their architecture. To simplify their use in Python, the "hdf5plugin" library (Vincent et al., 2022) has been developed to provide an accessible high-level interface that translates user-defined options into parameters required by the

HDF5 compression filters, making them usable from the "h5py" library (Collette, 2013), the HDF5 Python interface. While the "hdf5plugin" library successfully bridges the gap between the h5py Python interface and the HDF5 compression filters, its application necessitates significant code modifications, particularly for users accustomed to working with higher-level libraries such as xarray, who might find these adjustments less intuitive and more complex.

In the approach we present here, we add another layer and propose a user-friendly "compression specification format" (CSF)

that allows compression specifications to be expressed as simple, plain text, making the use of compression more accessible to users. We developed a method to translate the textual compression specifications into the numerical parameters expected by the HDF5 compression filters. The goal is to make "hdf5plugin" applicable beyond "h5py" to Python libraries including "h5netcdf"



(h5netcdf Contributors, 2023) and "xarray" (Hoyer and Hamman, 2017), which are extensively used in the atmospheric science community.

The CSF is a string of comma-separated values, which include the type of compression, the compressor, mode, and parameters. While formats including YAML (Ben-Kiki et al., 2009) and JSON (Bray, 2017) can be more verbose and carry more information, we decided for comma-separated strings to obtain a format that is easy to understand by humans and that facilitates straightforward use from command lines as well as from shell scripts or Fortran namelists, typically used in numerical weather prediction. For lossless compression, specification of compressor, mode, and parameters can be omitted to use the
default values chosen by BLOSC.

The CSF then takes the format:

```
"lossy,zfp,accuracy,0.01", or "lossy,sz,abs,0.01", or "lossless"
```

The strings can be extended to include different specifications for different variables in the same line. Here, the name of a variable is followed by a colon (:), and spaces separate multiple specifications:

```
"temperature:lossy,sz,abs,0.01 precipitation:lossy,sz,pw_rel,0.0001"
```

If a specification is provided for a single variable only, all other variables in a data file will be compressed lossless by default:

```
"temperature:lossy,sz,abs,0.01"
```

Default specifications can be defined explicitly. The previous specification is equivalent to:

```
"temperature:lossy,sz,abs,0.01 default:lossless"
```

However, it is possible to specify arbitrary defaults:

```
"temperature:lossy,sz,abs,0.01 default:lossy,zfp,rate,6.4"
```

Coordinate variables are by default always compressed lossless. To change this behavior, the "coordinates" keyword can be used:

```
"coordinates:lossy,zfp,rate,8"
```

## 2.3  Inferring optimal compression parameters from user-specified quality metrics

For further processing and analyzing the compressed data, it may be desirable to ensure that a specific quality metric that may be important for a given use case is maintained under compression. Such quality metrics can include Pearson's correlation and the mean square error (MSE) and its variants the root mean square error (RMSE), and the normalized root mean square error (NRMSE). Other more specific metrics include the Structural Similarity Index Metric (SSIM; Wang et al., 2004) commonly
used in the computer vision literature, and the Continuous Ranked Probability Score (CRPS; Gneiting et al., 2005).



In our approach we include a method to automatically find optimal compression parameters based on specified quality metrics. We build on the work by Kunkel et al. (2017) and Tao et al. (2019b). Kunkel et al. (2017) introduces the idea of decoupling the compressor selection and mode from the quality constraints. They allow the user to specify one of six predefined metrics and automatically find the best-performing compressor. In Tao et al. (2019b), a similar approach is used, focusing on making the selection on the fly. Our approach also tries to optimize compression specifications only based on quality constraints but making it extendable to any metric of interest.

Two possibilities exist when trying to optimize compression parameters: (1) maximizing the compression ratio while maintaining specific quality metrics, and (2) maximizing quality metrics while achieving a specific compression ratio. The solution in either scenario comprises a selection of a compressor, a compression mode, and a parameter. Since all the compression methods used in this work are uni-parametric, the optimal parameter search reduces to a 1-D optimization problem. Various methods are available to solve 1-D optimization problems (Kochenderfer and Wheeler, 2019); the bisection method (Burden et al., 2015) is used in our work. Figure 1 shows an illustration of the steps this method would do in order to find optimal compression parameters.

Given a compressor and a method, we determine the parameter range, i.e., the relative error can go from 0 to 1. Then, we define a function that, given the compression parameter, returns the values of the metrics of interest. Then, we use the bisection method with this function to find the optimal parameter. The user can select the metrics of interest. Pearson's correlation, MSE, RMSE, NRMSE, SSIM, and CRPS have been implemented. If more than one metric is used, the algorithm will find the parameter that fulfills all the constraints. If no compressor or method is provided, the search will be performed for all of them and the best performing one will be selected.

## 2.4 Integrating HDF5 compression filters in "enstools"

We integrated our approach in the input/output (I/O) implementation of the ensemble tools ("enstools") Python package (Redl et al., 2022), a collection of weather research utilities that include methods for clustering, interpolation, I/O, access to remote open data, post-processing, and evaluation scores. In addition, we provide a lightweight stand-alone Python package, "enstools-encoding" (Tintó-Prims, 2022b), to simplify adoption of our developments without the need to use the entire "enstools" package. The I/O implementation of "enstools" is based on "xarray", hence further tools that rely on xarray for I/O can adopt our approach without adding much complexity.

"enstools-encoding" processes the CSF string and assigns a compressor, mode, and parameter to each variable. Next, these specifications are converted to the actual parameters required by the HDF5 compression filters, including a compression filter identification number and an array of options. Because different filters use different options, it is necessary to implement an interface for each compressor. The complexity of the interfaces depends on the specific filter but typically at most ten lines of code are required. Our implementation is based on hdf5plugin (Vincent et al., 2022) that already includes several lossless compressors and ZFP. By adding a new interface, we extended the existing implementation to also provide access to SZ.

Additionally, we provide a command line interface, "enstools-compression" (Tintó-Prims, 2022a), to compress existing datasets without the need to code Python scripts. It uses "enstools" to read files, which in addition to using NetCDF and HDF5





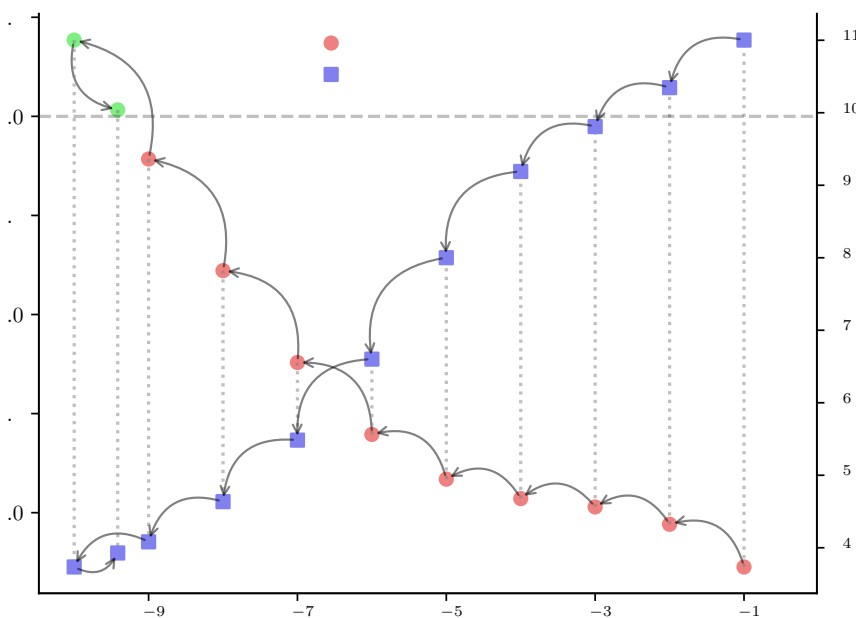

**Figure 1.** Illustration of how the bisection method works to find the optimal parameter. The circles represent the correlation indices corresponding to each parameter. The arrows represent a step of the algorithm. In this case, we are using SZ with the relative error mode. Since, in this case, the parameter range goes from 0 to 1, we perform the first evaluation at the middle of this range and keep adjusting the parameter until the value of the correlation index is within a defined threshold (indicated with a dotted horizontal line). The variable temperature at 2 meters above the surface from the ERA5 dataset was used.

also allows processing of GRIB files. and writes them again using the desired compression specification as HDF5 files. The tool provides additional features including keeping only certain variables, emulating the compression without writing output files, or parallelizing the compression of multiple files on a cluster.

## 3 Applications

In this section, a sequence of use cases will be presented, where lossy compression has been applied in different research applications. The aim is to demonstrate through examples how compression settings can be chosen, what level of compression is possible without compromising the scientific results, and to give some indications of where a researcher must be careful in applying lossy compression. The examples are all based on recent studies carried out within the Waves to Weather research program (Craig et al., 2021). Section 3.1 discusses the achievable compression ratios for standard model output data when specific quality constraints are applied. A representative set of variables is drawn from the ERA5 dataset, and the tradeoff between



the degree of compression and various quality metrics is explored. While this assessment provides some useful guidelines, it
does not necessarily give confidence to scientists who work with more sophisticated diagnostics that are derived from the compressed data. Therefore, the subsequent examples will consider some more advanced use cases. It is impossible to cover all the possible applications of atmospheric data, but using range of different studies, some indication can be given of the benefits and pitfalls that may be encountered. The next example in Section 3.2 considers the forecast error growth experiments of Selz et al.

(2022), and highlights a difficult scenario where the important signal is a small difference between large values. Section 3.3, based on the forecast evaluation study of Matsunobu et al. (2022) uses a complex verification metric where it is difficult to predict in advance what the effects of compression will be. The final example, or rather set of examples, in Section 3.4 uses the visualization tool Met.3D (Rautenhaus et al., 2015b). Examples drawn from Rautenhaus et al. (2020) are used to explore how changes in compression levels can alter the visual interpretation of meteorological data.

## 240   3.1   Compressing ERA5

The choice of optimal compression parameters depends on the properties of the variable being compressed, and on the application that the data will be used for. Often a data is used for a wide range of applications, which are not necessarily known in advance. In such cases, it is most reasonable to use relatively simple error measures to evaluate the quality of the compressed data set, but to require a high threshold of accuracy, i.e. to be conservative in discarding information. In this example, we con-

sider the ERA5 data set (Hersbach et al., 2018). Since ERA5 is widely used in weather and climate studies, the compression specifications must not be specific to a specific application. The quality criteria used here for the compressed data set will be based on previous studies, and we will investigate the degree of compression that is possible for different variables using different compression algorithms. Finally, we will show an example with overly aggressive compression to provide a visual impression of the compression artifacts that we want to avoid.

When compressing ERA5 data, we will require that the correlation with the original field be at least 0.99999. As discussed in the introduction, atmospheric data contains uncertainty and can be reduced without necessarily affecting the real information content. A correlation of 0.99999 has been mentioned multiple times in the literature as a threshold (for example by Tao et al. (2019a)), and Baker et al. (2014) suggests that structural similarity can be a useful additional metric. Therefore, we will also require a structural similarity index metric higher than 0.99. The search algorithm (see 2) implemented in "enstools" (Redl

et al., 2022) is able to search for compression parameters that simultaneously guarantee multiple quality metrics, and this will be applied to a representative sample of ERA5 variables (listed in Table 1). Of course there is no guarantee that the quality metrics shown here are adequate for every application, but they should be an useful starting point to see what compression ratios can be expected for different variables when these quality constraints are applied.

Figure  2 shows the compression ratios that are obtained when applying lossy compression to different variables using differ-

ent compressors and methods. Here, the compression parameters are selected ensuring that the data that has been compressed has at least a correlation of 0.99999 and a structural similarity of at least 0.99 with the original data. The degree of compression that is possible even with these conservative criteria is substantial, ranging from a maximum of 90.8 times for the geopotential to 5.5 times for the vertical velocity. As might be expected, larger reductions are possible for smoother fields. Interestingly,



| Long name | Short name |
|---|---|
| Divergence | d |
| Fraction of cloud cover | cc |
| Geopotential | z |
| Ozone mass mixing ratio | o3 |
| Potential vorticity | pv |
| Relative humidity | r |
| Specific cloud ice water content | ciwc |
| Specific cloud liquid water content | clwc |
| Specific humidity | q |
| Specific rain water content | crwc |
| Specific snow water content | cswc |
| Temperature | t |
| U component of wind | u |
| V component of wind | v |
| Vertical velocity | w |
| Vorticity | vo |

**Table 1.** List of ERA5 3-D variables used in Figure 2. The table provides a correspondence between the short names used in the figure and their respective full descriptions.

the most performant method is not the same for all variables. However, we can see that SZ outperforms ZFP for most of the

variables in the dataset.

A particularly challenging problem is to know when the data has been over-compressed and too much information is lost, without access to the uncompressed data for comparison. A rough idea can be obtained by looking for unphysical artifacts in images of the field. Figure 3 shows a comparison between non-compressed and compressed total column water. With the quality criteria applied in Figure 2, the differences would be invisible at this resolution, so an example is shown with much

stronger compression. In particular, SZ3 allowing a relative error of 5% was used, giving a compression ratio of 241.5. Even with this degree of compression, it is difficult to see differences on the full images, but if we look at the zoomed section artifacts are more evident. They take the form of patches of enhanced gradients that are aligned with the coordinate directions, and as a result, are easy to identify as unphysical.

## 3.2 Error growth analysis

Although a compression level that maintains a correlation of > 0.99999 seems sufficient for most meteorological applications, especially given the inherent uncertainty of atmospheric data, there are exceptions where much higher correlations or even lossless compressed data is needed. One example is studies that investigate the intrinsic limit of predictability by applying





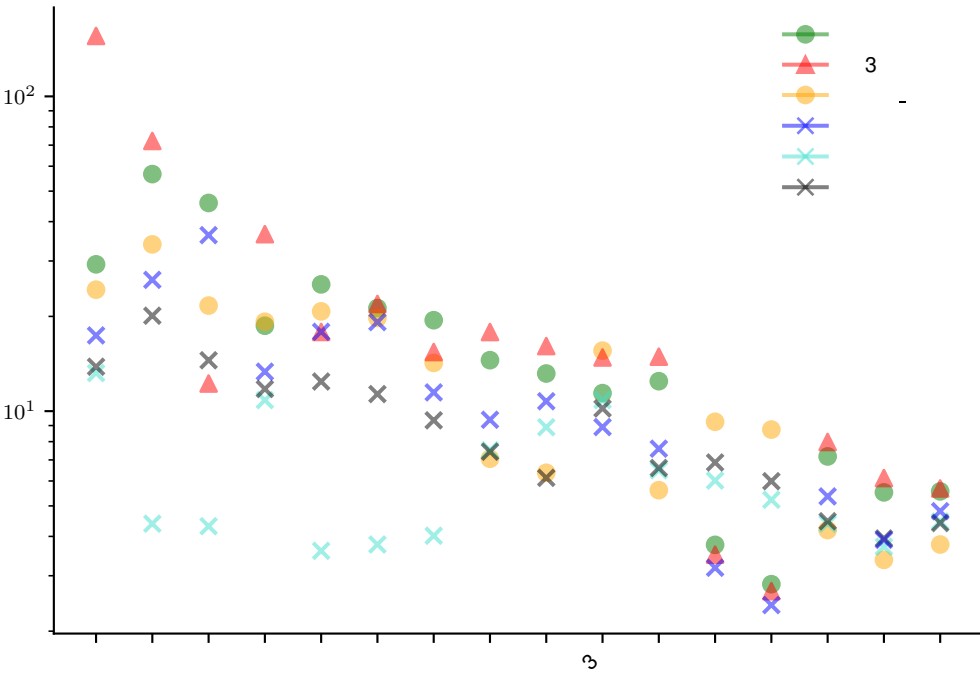

**Figure 2.** Compression ratios and bit rates for different ERA5 three-dimensional variables listed in Table 1. Compression parameters are found keeping a correlation of at least 0.99999 and an SSIM of at least 0.99.

a perfect model assumption and starting simulations from very small differences in the initial conditions (sometimes called identical twin experiments). We demonstrate the effects of lossy compression for such a case based on data from our earlier

predictability study (Selz et al., 2022). Fig. 4 shows the difference in kinetic energy (DKE) at 300hPa for the initial time and different lead times early in the forecast (see Selz et al. (2022) for a detailed explanation of the experimental design). The DKE has been calculated twice, from the uncompressed float32 output data and from data that has been lossy compressed to 8 bits per value (factor 4) using ZFP. Both DKE results are shown in the figure and the error is shaded. It can be seen that if the initial condition uncertainty is reduced to 10% with respect to current levels, some significant errors already occur at the smallest

scales and at the initial time. However, in the experiment where the initial condition uncertainty has been further reduced down to 0.1%, the compressed data fails to capture the relevant information entirely and the DKE spectrum at the initial time purely consists of noise from the compression algorithm. In both cases, the errors are negligible at later forecast lead times (>24h), and also in the background spectrum. This example demonstrates that the compression rates must be selected carefully if diagnostics are considered that involve differences from very similar values.



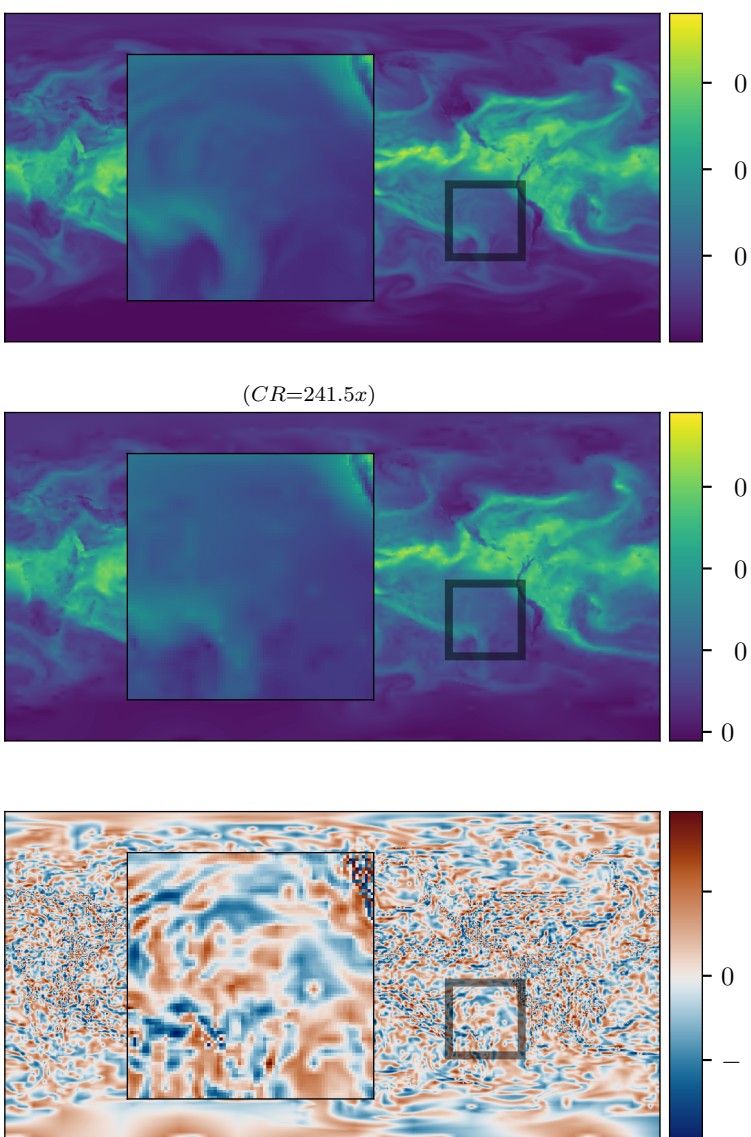

**Figure 3.** Total Column Water from ERA5. In panel (a) we can see the uncompressed reference, panel (b) shows the data that has been compressed and panel (c) shows the difference between both. The zoom square shows the region with higher differences. The data has been compressed using SZ3 with a relative error method with a threshold of 5%, leading to a compression ratio of 241.5x i.r.t. single precision.





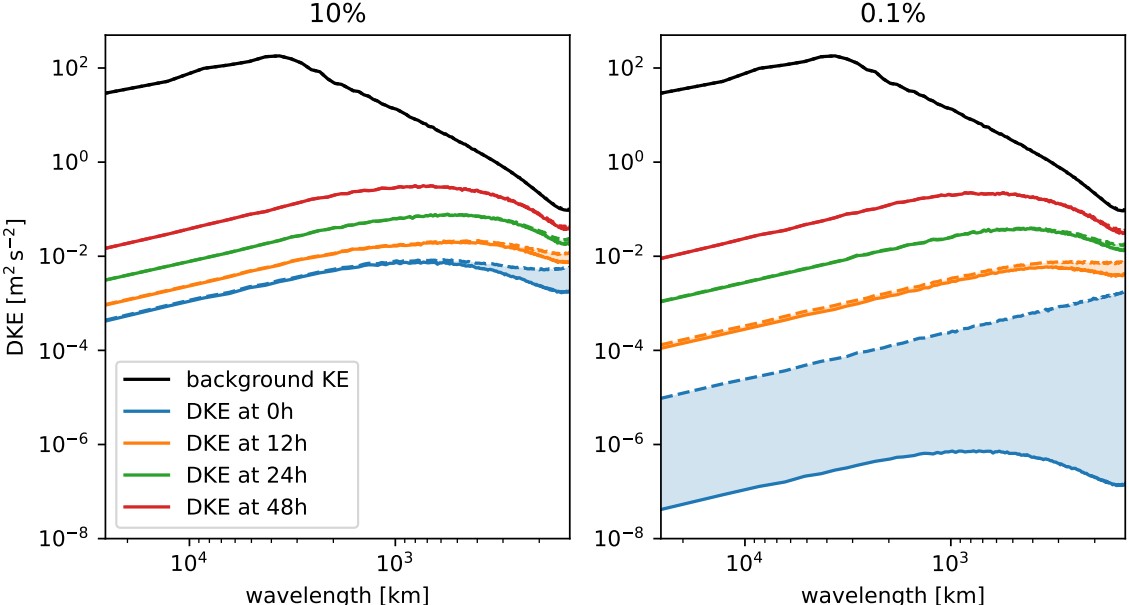

**Figure 4.** Spectra of Difference Kinetic Energy (DKE) from initial condition sensitivity experiments (Selz et al., 2022). Solid lines show the results from uncompressed data. Dashed lines show the result from compressed data. The shaded areas highlight the errors introduced by the compression.

## 3.3  Fraction Skill Scores

The next example considers a forecast skill score derived from the raw model output through a series of transformations that include a highly nonlinear threshold, as well as information that is nonlocal in space. The fractions skill score (FSS; Roberts and Lean, 2008) is a diagnostic tool used in many national weather services to evaluate a spatial forecast skill of binary fields, for example the presence of precipitation exceeding a certain threshold. At each location, FSS counts the fraction of grid points with precipitation in a rectangular neighborhood centred at that point, and compares it to corresponding fraction from a second data set (e.g. forecast vs observation). The results for the individual locations are then averaged in space. The size of the rectangular neighborhood for which FSS exceeds a specified value can be used as a maximum scale of good agreement. That scale is named the believable scale and can be used to represent the time evolution of spatial variability (Dey et al., 2014; Bachmann et al., 2018, 2019; Matsunobu et al., 2024). To compute the FSS from total precipitation, we first need to convert it to a binary field. We do that using a threshold, which is defined here using a percentile value of the field to keep the overall number of precipitating grid points the same between the compared fields. Because we are using a threshold, small changes in the precipitation amounts can lead to differences in the binary fields that might be amplified to distinctive differences in FSS and believable scales.

Here we use FSS and believable scales to diagnose how far ensemble precipitation forecasts have diverged from each other, reproducing the work presented in Matsunobu et al. (2022). In this case, the believable scale can be interpreted as





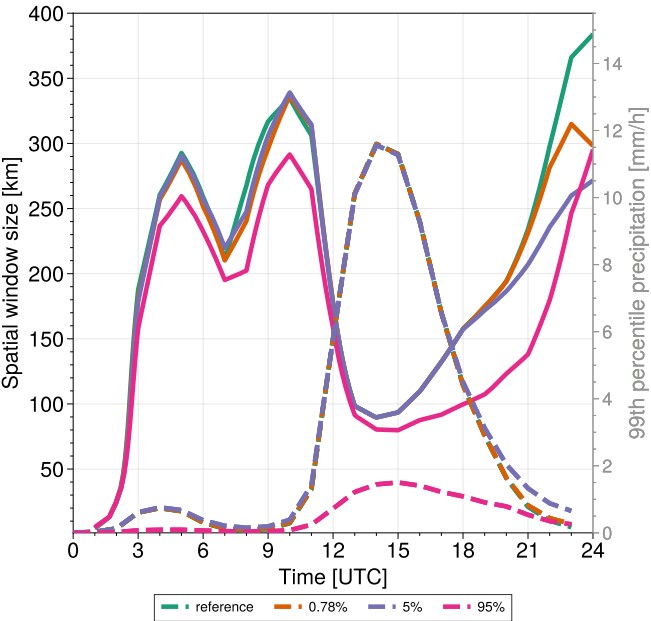

**Figure 5.** Believable scales for hourly precipitation. Solid lines show the temporal evolution of believable scale. Dashed lines show the threshold values used for binarization. Green lines were produced with the reference data, the orange lines were produced with data that was compressed using SZ with a point-wise relative error mode with a threshold of 0,78%, the purple line was produced with a point-wise relative error of 5%, and the pink with a point-wise relative error of 95%.

the scale where the ensemble members are similar, with precipitation features on smaller scales differing strongly between ensemble members. Fig. 5 shows the believable scale as function of forecast lead time for different levels of compression of the underlying precipitation fields. A systematic reduction of the believable scale is found for all compression levels. The effect is more noticeable during the periods from 6-9 UTC and 20-24 UTC, when precipitation is weak and the threshold for

310 binarization is low. At these times, small differences introduced by the compression can have large effects on the number of points exceeding the threshold, changing the FSS and the believable scale. It should be noted however, that the exact values of FSS and believable scale at such times may not be of great importance, since our focus tends to be on the time of large precipitation amounts.

Interestingly, the believable scale remains qualitatively similar even for compression that allows a point-wise relative error

315 of 95%. This is true even though other statistics such as the 99th percentile precipitation amount has completely lost accuracy (dashed lines in Fig. 5). This indicates that the tolerance of FSS to information loss is high. However, we should also keep an eye on the threshold to avoid misinterpretations. Overall, reasonable amounts of lossy compression does not affect a scientific conclusion drawn by FSS analyses, although the effective compression level should be carefully assessed. In this example, a point-wise error tolerance of 5% still satisfies the required accuracy for both the believable scale and 99th percentile value.



### 3.4 Interactive 3-D Visualization

Visualization is an important and ubiquitous tool in the daily work of atmospheric researchers and operational weather fore-casters to gain insight into meteorological simulation and observation data (see overview by Rautenhaus et al., 2018). In our context, the question arises to which extent visualizations that are used for data analysis or communication purposes are im-pacted by usage of compressed data: Would a scientist or forecaster still draw the same conclusions from the image, or will artifacts be introduced that will cause misleading conclusions? Of course, the range of possible visual depictions of meteorolog-ical data is large and ranges from 2-D maps over 3-D depictions to application-specific diagrams such as ensemble meteograms (Rautenhaus et al., 2018). The extent to which these depictions will be impacted by compression can also be expected to cover a wide range.

In visualization science, compression has been much discussed in the context of interactive large data rendering (see overviews by Balsa Rodríguez et al., 2014; Beyer et al., 2015). The main challenges are the bottleneck of reading data from disk into computer memory fast enough for interactive visualization, and fitting the data into the memory of graphical process-ing units (GPUs) while retaining properties required for visualization algorithms, including random access if the data is kept in compressed form in GPU memory, local decompression (i.e., being able to decompress subsets of the data only), and real-time demands for decompression during rendering. Despite recent advances in graphics hardware including increasing memory sizes (at the time of writing, GPUs ship with up to 24-48 GB of video memory), the increasing data volumes output by (not only meteorological) simulation and observation system keep these issues salient. Recent related research includes studies on the trade-off between precision and resolution on visualization when reducing data volumes (Hoang et al., 2019, 2021), development of new compression schemes that fulfil specific application-motivated criteria (e.g., preserving features such as critical points in vector fields; Liang et al., 2020), and, following the recent advances in machine learning, investigations of the suitability of neural-network-based compression schemes for rendering (e.g., Lu et al., 2021; Weiss et al., 2022).

Here we consider the example of combined 2-D–3-D depictions as generated in interactive visual analysis workflows, and in particular, by the meteorological visualization framework Met.3D (Rautenhaus et al., 2015b). Comparable software options include Vapor (that also natively supports lossy wavelet compression; Norton and Clyne, 2012; Li et al., 2019) and ParaView (Ayachit et al., 2012), cf. the overview in Rautenhaus et al. (2018). Met.3D has been used, for instance, for weather forecasting during atmospheric field campaigns, including the 2016 NAWDEX campaign (Schäfler et al., 2018, e.g. their Fig. SB2). Such forecasting requires forecast data, as soon as it is available, to be transferred at minimum time from a weather centre (e.g., ECMWF) to a visualization server. In particular if 3-D ensemble forecast data is required (note that 3-D visual analysis requires best possible vertical resolution, i.e., all available model levels, cf. Rautenhaus et al. (2015a) for an example case), datasets generated by current forecast systems encompass multiple 100 GB or more at the time of writing. Internet bandwidth is a limiting factor for this application, and data compression can make the difference between being able to use the latest forecast for forecasting or not.

For illustration, we recreate two visualizations with different characteristics that were generated during a demonstration session of 3-D visual analysis for forecasting during the 2019 Cyclone Workshop. All figures have been created from model





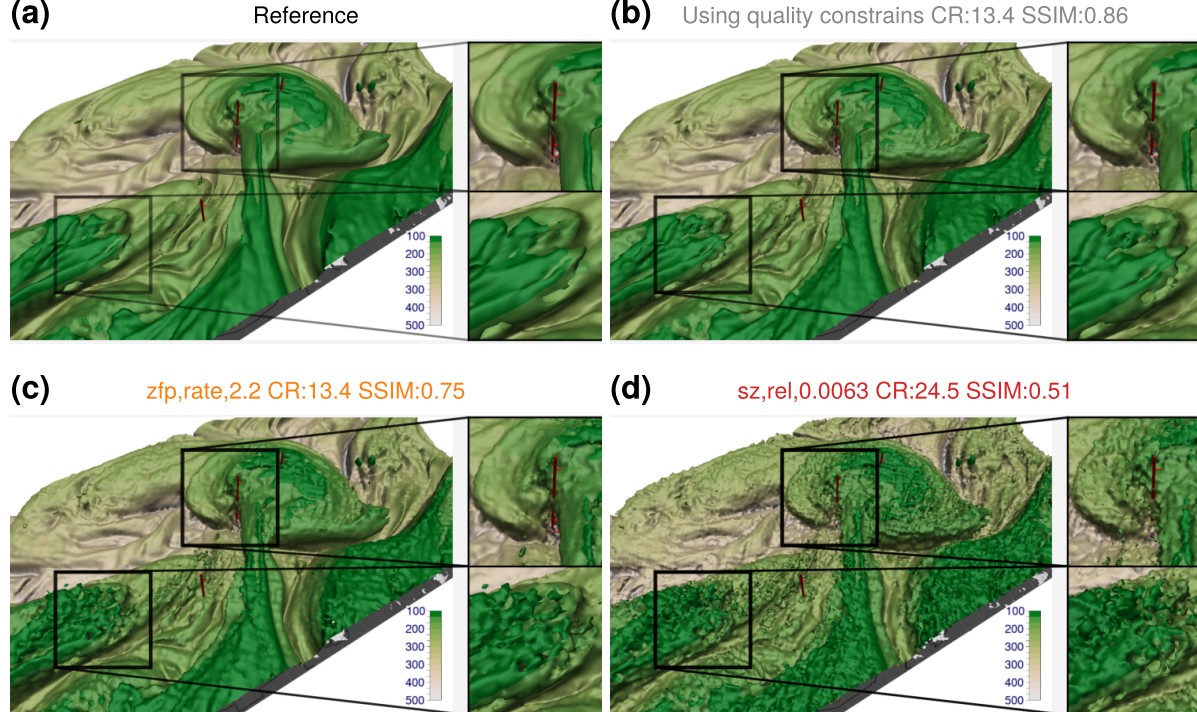

**Figure 6.** Impact of lossy compression on a 3-D visualization of the dynamical tropopause as represented by the 2-PVU potential vorticity (PV) isosurface, with PV computed from the 3-D NWP variables horizontal wind and temperature. Reference figure (a) reproduced from Rautenhaus et al. (2020), showing member 24 of the ECMWF ensemble forecast from 00 UTC 29 September 2019, valid at 00 UTC 03 October 2019. Colour coding shows pressure elevation (hPa). (b) Quality constraints of correlation index = 0.99999 and SSIM = 0.99 imposed on the NWP variables lead to a compression ratio (CR) of 12, the resulting visualization has a SSIM of 0.89 compared to the reference. (c-d) Visualizations produced from the NWP variables compressed with further settings, illustrating artefacts introduced by the compressors. Captions list compression parameters, achieved compression ratio, and SSIM of the resulting image.

level data from the ECMWF ensemble prediction system, interpolated to a regular latitude-longitude grid with a grid spacing

of 0.5° in both dimensions. The original depictions can be found in Rautenhaus et al. (2020). Figure 6 shows a 3-D depiction of the dynamic tropopause as represented by the 2-PVU isosurface of potential vorticity (PV). PV has been computed as a derived variable from compressed fields of horizontal wind and potential temperature, using the implementation contained in the LAGRANTO package (Sprenger and Wernli, 2015). Note that PV computation involves derivatives; for visualizations depicting variables computed from derivatives of compressed variables, we expect a stronger impact of compression on the

visual result than for visualizations depicting only the underlying compressed variables directly (also cf. the discussion in Hoang et al., 2019).

     We first compress the model variables with the quality constraints discussed in Sect. 3.1 (i.e., enforcing a correlation of 0.99999 and SSIM of 0.99 between original and compressed data fields), using the automatic optimization of compression





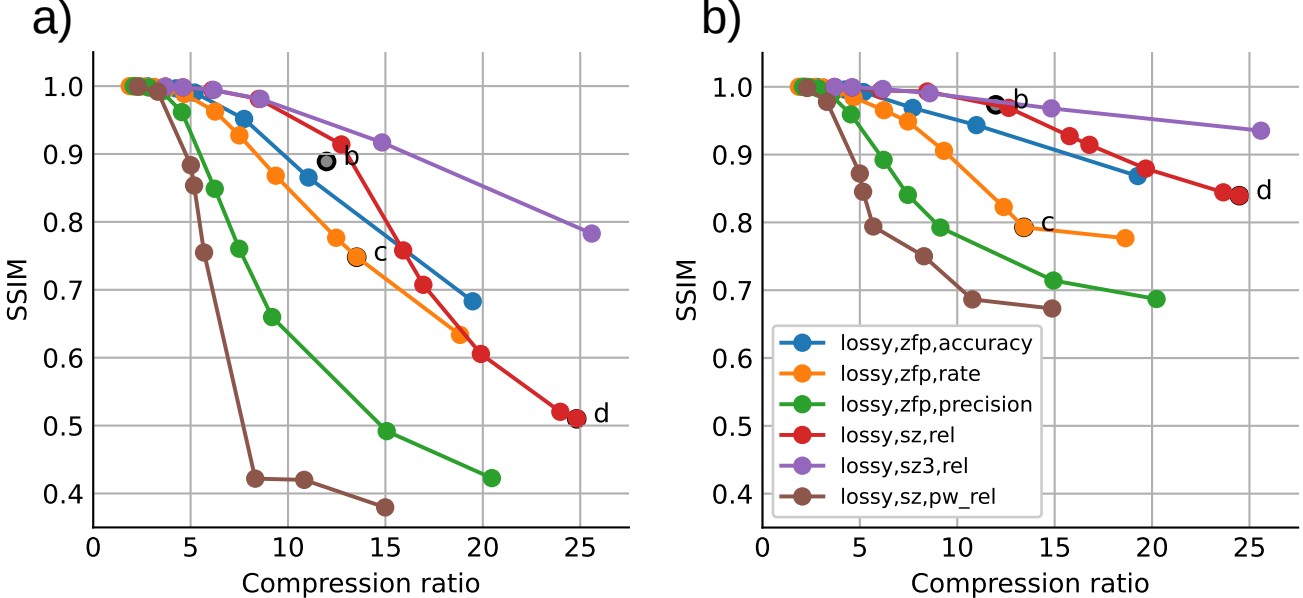

**Figure 7.** Dependence of the visualization SSIM (i.e., the SSIM computed between the visualization image generated from uncompressed data and that generated from compressed data) of (a) Fig. 6 and (b) Fig. 8 on compression ratios, for different compressors. Dots with a black outline correspond to panels b-d in Fig. 6 and Fig. 8.

parameters discussed in Sect. 2.3. Note that this approach leads to different compression parameters for each of the model

variables, from which PV is subsequently derived. The resulting visualization achieves a SSIM of 0.89 (Fig. 6b; we also use the SSIM metric to compare the visualizations generated from compressed data to a reference generated from uncompressed data; following Wang et al. (2004)). This leads to small but noticeable differences in the isosurface structure in the enlarged regions of the image. However, given the achieved overall compression ratio of 12, the image in its original extent shows only few noticeable visual artefacts, and we argue that an analyst would likely draw the same conclusion than from the reference

visualization (Fig. 6a). Note that this statement is subjective and reflects the authors' view. Our objective for this article is to provide the reader with an idea of how artefacts introduced by lossy compression look like for typical meteorological 3-D visualizations. A much more detailed study on the differences that users perceive would need to be carried out to obtain results about "best acceptable compression settings" for visualization purposes.

Compressing each model variable with the individual compression parameters inferred from quality constraints may be

undesirable, e.g., if predictable file sizes and hence a specified compression ratio is required. We hence compare how the visualization is impacted if generated from data sets in which each model variable has been compressed with the same compression parameters. Figure 7a shows how the SSIM of Fig. 6 decreases as compression ratios increase for different compression parameters of the sz and zfp compressors. The figure clearly shows how different the different compressors perform for the selected



example. While the visualizations generated from data compressed with sz in point-wise relative mode quickly deteriorate in
quality and the visualization's SSIM drops below 0.7 at a compression ratio of about 7, visualizations generated from data
compressed with sz3 in "global" relative mode maintain an SSIM of over 0.8 for compression ratios exceeding 20. The zfp
compressor performs in between.

Interestingly, the sz compressor (in particular in the sz3 version) achieves even higher visualization SSIMs for the same
compression ratio compared to the quality-constraint-compressed dataset in Fig. 6b. While in general we consider it desirable
to achieve pre-defined error metrics for each variable's data (as done in Sect. 3.1 and Fig. 6b), for the considered type of
visualization, compressing each variable with the same compression parameters leads to fewer visualization artefacts. We
hypothesize that non-linear effects may increase the error for derived fields – PV is computed from horizontal wind and
temperature, and the errors in these fields may be distributed differently when using different compression parameters.

For illustration of artefacts introduced by the compressors, Fig. 6c and d show examples of visualizations that achieve SSIMs
of 0.75 (using the zfp compressor with a compression ratio of 13.4) and 0.5 (using the sz compressor with a compression ratio
of 24.5). As Fig. 6c shows, the large-scale structure of the PV isosurface is still clearly discernible, however, when enlarging
parts of the image, clearly noticeable artefacts occur. In Fig. 6d, the visual artefacts that have been introduced very negatively
impact visual analysis of features of the scale of the enlarged regions.

As a second example, Fig. 8 shows a combination of 2-D displays (contour lines, colour coding), as well as jet-stream core-
lines as an example of a visual abstraction of an atmospheric feature relevant for the analysis (also reproduced from Rautenhaus
et al., 2020). The core-lines have been computed using the method described by Kern et al. (2018). They also heavily rely on
derivatives, here, derivatives are computed from compressed data of the three wind field components (of which horizontal wind
speed is also colour coded on the vertical section). The plots in Fig. 8 are organized in the same way as those in Fig. 6, using the
same compression parameters. Similarly, Fig. 7b shows how the figure's SSIM changes with changing compression parameters.
Since in this case the overall visualization contains larger parts that are not influenced by compression of the model variables
(mainly the base map and parts of the vertical section where wind speeds are below the range of the colour map), the resulting
SSIMs are higher compared to Fig. 6. For the automatically determined compression parameters in Fig. 8b, an SSIM of 0.97
is achieved. The resulting visualization looks very much like the reference, while being based on data compressed with a ratio
of 12. For the contour lines of mean sea level pressure and potential temperature, and the colour coding of wind speed, we
attribute the high similarity to the fact that these visual representations have been produced directly from the model variables
with no derivative computation involved. The jet-stream core lines, however, are –in addition to being based on derivatives–
also sensitive to wind speed thresholds and further filter parameters (Kern et al., 2018), hence we expect a larger sensitivity to
compression. In fact, in the upper enlarged region, one of the shorter arrows close to the hurricane center is missing in Fig. 8b.
When using data compressed as in Fig. 6c, visible differences occur in the detected jet cores, and the zfp compressor also
introduces blocking artefacts in the mean sea level pressure contours. When increasing the compression ratio to 25 for the sz
compressor (as in Fig. 6d), the sensitivity of the jet cores becomes most apparent – many lines from the reference are missing.

Of course, Figs. 6–8 only illustrate the impact that lossy compression can have on visualizations of two very specific cases.
We selected the examples to highlight the impact on both typical 2-D visualizations elements (including contour lines and





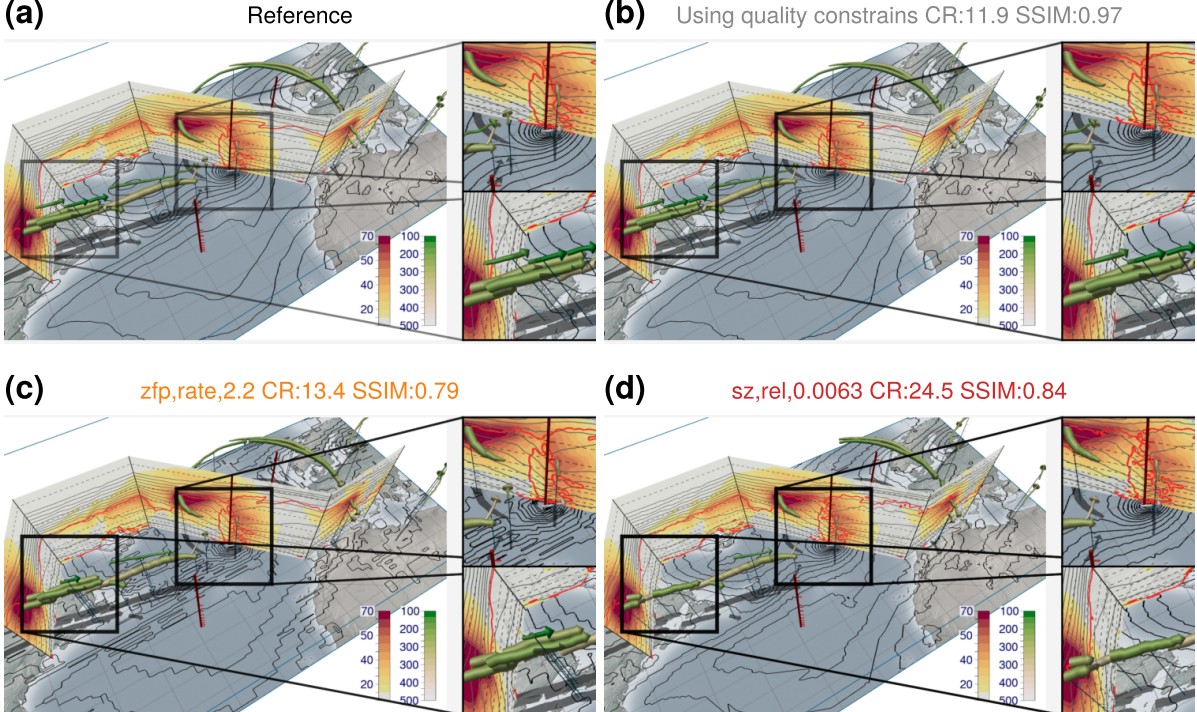

**Figure 8.** Same as Fig. 6 but for a visualization showing jet-stream core-lines computed from the horizontal wind field (green tubes, colour shows pressure elevation in hPa), a vertical section (colour shows horizontal wind speed in ms$^{-1}$, 2-PVU contour in red, potential temperature contours in grey), and a surface map with contours of mean sea level pressure (black contours). Reference figure (a) reproduced from Rautenhaus et al. (2020). Note the different SSIMs compared to Fig. 6.

colour coding), the impact on derived variables (including PV), and the impact on advanced 3-D visualization elements (in-

cluding derived features such as jet-stream core-lines). As noted above, a much more detailed study, also including the aspect of precision vs. resolution (Hoang et al., 2019), will be necessary to yield more generic results on which data reduction approaches are "acceptable" for analysis and communication by means of visualization. However, it will be straightforward using our approach to conduct more comprehensive statistical analysis on further visual displays, and we encourage the reader to use our software to perform corresponding analyses for their own depictions.

# 4  Conclusions

The aim of this study has been to demonstrate the practicality and effectiveness of lossy compression techniques for meteorological and Earth sciences data. The 'enstools-compression' Python tool offers a seamless way to integrate lossy compression into NetCDF workflows, through its use of the HDF5 compression filter architecture. Advanced compression algorithms like



ZFP and SZ (2.x and 3) are employed, and tools and methods are have been developed to search automatically for compression
parameter settings that satisfy given requirements for data reduction and accuracy.

The results of this study are promising, showing that substantial data reduction can be achieved without compromising the
overall integrity and usability of the data for scientific analysis. Through the utilization of advanced compression algorithms
like ZFP and SZ (2.x and 3), the study has illustrated the potential for efficient data management while maintaining a high level
of data accuracy. The use of lossy compression has been illustrated with series of examples from current atmospheric science
research projects. Significant compression was found to be possible in all cases without compromising the scientific utility of
the data. In most cases, simply choosing a high level of accuracy, such as a correlation of 0.99999 and a structural similarity
of at least 0.99 with the original data, resulted in compression ratios of 5-100, without changing diagnostics derived from the
compressed data significantly. Nevertheless, exceptions like the example of error growth calculations in Section 3.2, show that
it is always necessary to think about how the data will be used.

It will be a significant benefit to science if data volumes can be significantly reduced while yielding the same results. We
hope that tools presented, with the addition of the example use cases will encourage and enable this.

*Code and data availability.* The software developed for this study, enstools-compression 2023.11, is publicly available and can be accessed
through its GitHub repository (https://github.com/wavestoweather/enstools-compression) and its corresponding zenodo entry (https://doi.
org/10.5281/zenodo.10998676). The code has an Apache-2.0 licence. This repository contains all the necessary documentation on how to
install, configure, and use the software.

The software to reproduce the experiments and produce the figures can be found in https://doi.org/10.5281/zenodo.10998604 .

The data analyses conducted in this study primarily utilize the ERA5 reanalysis dataset. The ERA5 data is not hosted by us but can be
accessed through the Copernicus Climate Change Service (C3S) Climate Data Store (CDS). Users interested in accessing the ERA5 dataset
can register and download the data by following the guidelines provided at the CDS website: https://cds.climate.copernicus.eu/. The ERA5
dataset is provided under specific terms of use, detailed on the CDS website, which users are encouraged to review before accessing the data.



**Appendix A: Usage**

We provide selected examples of the use of the Python utilities we provide along with this article. Detailed information can be found in the documentation (enstools-compression Contributors, 2023).

Using the command line interface, an existing NetCDF file can be compressed in the following way:

```
enstools-compression compress input.nc -o output.nc -compression lossy,sz,rel,0.0001
```

The same can be achieved with an "xarray" dataset present in a Python script:

```
enstools.io.write(dataset, "output.nc", compression="lossy,sz,rel,0.0001")
```

The resulting files can be read by any software capable of reading NetCDF using an HDF5 capable backend, as long as the corresponding HDF5 compression filter is available in the software.

To determine optimal compression parameters from quality metrics, the "analyze" keyword is passed to the command line

interface:

```
enstools-compression analyze input.nc
```

If neither compressor nor method are provided, the tool considers all implemented compressors and methods and determines the best-performing one. If the user does not specify the quality metrics that need to be kept, the default of a correlation of at least 0.99999 and a structural similarity of at least 0.99 are used. The command returns the resulting compression specification for all variables in the dataset.

The code architecture allows the definition of custom metrics by defining a Python function with the following signature:

```python
from xarray import DataArray
def custom_metric(reference: DataArray, target:DataArray) -> DataArray:
    non_temporal_dimensions = [d for d in reference.dims if d != "time"]
    return ((target - reference) ** 2).mean(dim=non_temporal_dimensions)
```

Within a Python script, this custom metric can be used as follows:

```python
# Register the function as a new score
import enstools.scores
enstools.scores.register_score(function=custom_metric, name="custom_metric")
analyze_files(file_paths=[input_path], constrains="custom_metric:5")
```

Custom metrics can also be used from the command line interface by storing the custom function in a file with the same name as the function:

```
enstools-compression analyze my_files_*.nc \
--constrains custom_metric:5 --plugins /path/to/custom_metric.py
```





Consider the example of compressing a dataset consisting of a large number of files. A possible approach could be to analyze a fraction of the files to determine a compression specification:

```
enstools-compression analyze my_list_of_files_*.nc
```

The full dataset can then be compressed using the parameters found in the analysis:

```
enstools-compression compress my_list_of_files_*.nc -o /path/to/output/folder
--compression temperature:lossy,sz,rel,0.001 default:lossless
```

Also, the analyze command can output the results directly in a YAML file.

```
enstools-compression analyze output.nc --output compression_specification.yaml
```

This file can later be directly used for subsequent compression:

```
enstools.io.write(dataset, "output.nc", compression="compression_specifation.yaml")
```



*Author contributions.* **Oriol Tintó Prims:** Main author. Contributed to all sections, lead developer of "enstools-compression". **George Craig:** Contributed to general discussion and document editing. **Takumi Matsunobu:** Led the development of Section 3.3. **Kameswar Rao Modali:** Involved in initial discussions and experimental work for Section 3.4. **Marc Rautenhaus:** Played a central role in discussions, revisions, and led development of Section 3.4. **Robert Redl:** Engaged in general discussions and paper review. Main developer of the "enstools". **Tobias Selz:** Led the development of Section 3.2.

*Competing interests.* The authors have declared that there are no competing interests.

*Acknowledgements.* The research leading to these results has been done within the subproject "Z2" of the Transregional Collaborative Research Center SFB / TRR 165 "Waves to Weather" (Craig et al., 2021), www.wavestoweather.de, funded by the German Research Foundation (DFG).

AI tools were employed to refine the manuscript's language and structure for enhanced clarity.



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
