# Peer review of "The effect of lossy compression of numerical weather prediction data on data analysis: a case study using enstools-compression 2023.11"

_EGUsphere, 2024_

## Author Response (AR1)

We would like to express our sincere gratitude to both reviewers for their thorough and insightful evaluations of our manuscript. Your constructive feedback has been invaluable in helping us improve the quality of our work, and we greatly appreciate the time and effort you dedicated to this review process. Below, we have provided our detailed responses to each of your comments and suggestions. We first address the points raised by Reviewer 1, followed by our responses to Reviewer 2.

**Reviewer 1:**

**Major Comments:**

1. **The proposed compression tool is used to reduce the size of, presumably, large datasets. One concern is the efficiency of the compression tool. As a perhaps extreme example, when I tried to use it with a 13G global ocean model output, the tool is very slow on my personal laptop. It would be instructive to indicate:**
2. **a) Are the compression algorithms pure Python, or are C/Fortran libraries used for compression?**

The compression algorithms employed are predominantly C++ (ZFP and SZ3) and C (SZ). The HDF5 filters, which serve as the interface between the compressors and the HDF5 library, are written in C or C++.

1. **b) Can we use the parallel capability in xarray to speed up the processing?**

Some degree of parallelism is possible, although in our experience, the behavior of xarray parallelism via Dask can be somewhat non-intuitive. However, embarrassingly parallel approaches, such as working with different files, do scale effectively.

1. **c) Can authors provide some comments on the efficiency of the tool?**

The tool is designed to achieve performance on par with the compressors themselves. Therefore, the efficiency should be comparable to that of the underlying compression libraries.

Additionally, we have incorporated this information into the manuscript to provide readers with more context regarding performance. Specifically, we added a note in the section discussing the compression schemes, stating that while the performance of the different compressors is outside the scope of this publication, detailed information on the performance of these compressors can be found in Zhao et al. (2020a). This ensures that readers are directed to relevant sources for a more in-depth understanding of the compressor performance.

1. **In Figure 1, 2, axis labels and legend labels are not shown. In Figure 3, the color bar does not have labels and values. It also does not have subfigure**

**labels such as a), b), and c) which are used in the caption. These issues make some results difficult to interpret.**

These issues were only visible after the editorial processing. We have made changes following the editor's advice to prevent this from happening in future versions. We hope that with these adjustments, the figures will display properly in the final publication.

**Minor Comments:**

1. **L95, page 4, it says '...this work aims to ensure that scientists can seamlessly utilize the compressed data. Essentially, the intent...'. Does the first sentence have the same meaning as the second sentence starting from 'Essentially'? Otherwise, I feel that the meaning of "seamlessly utilize the compressed data" is unclear.**

The first "seamless" refers to the decompression process, while the second "seamless" refers to the overall adoption of lossy compression. We have revised the second sentence to clarify this distinction:

*Original: "Additionally, this work aims to ensure that scientists can seamlessly utilize the compressed data. Essentially, the intent is to facilitate a seamless integration of lossy compression into the research workflows of the weather and Earth science communities."*

*Revised: "Additionally, this work aims to ensure that scientists can seamlessly utilize the compressed data. Essentially, the intent is to facilitate an effortless integration of lossy compression into the research workflows of the weather and Earth science communities."*

1. **L136, page 5, would it be better to use 'ranging from x_0 to x_1 (x_1 > x_0)' than specific values?**

We intend to use specific values to provide a concrete example of what would happen in a practical case. We believe this approach aids in comprehension, so we have opted to both use the symbolic notation suggested by the reviewer and to retain the specific values.

*Before: "To illustrate the difference between relative and point-wise-relative modes, consider a dataset with values ranging from 1 to 1001. When using the relative mode with a global relative error of 1%, the compressed dataset will contain errors smaller than $(1001 - 1) \cdot 0.01 = 10$. When using the point-wise-relative mode with the same threshold, each data point in the compressed dataset will have an error smaller than 1% of its original value. Hence, values at the higher end of the range will have absolute errors smaller than 10, and those at the lower end will have absolute errors smaller than 0.01. Note that in the example a global absolute error of 10 can lead to small positive values less than 10 becoming negative in the compressed representation. Hence, the point-wise-relative mode must be used when data values must remain strictly positive."*

*After: "To illustrate the difference between relative and point-wise-relative modes, consider a dataset with values ranging from $x0 x\_0 x0$ to $x1 x\_1 x1$ (where $x1 > x0 x\_1 > x\_0 x1 > x0$). When using the relative mode with a global relative error of $\epsilon \epsilon$, the compressed dataset will contain errors smaller than $(x1 - x0) \cdot \epsilon (x\_1 - x\_0) \cdot \epsilon (x1 - x0) \cdot \epsilon$. For example, if*

*x0=1x_0 = 1x0=1, x1=1001x_1 = 1001x1=1001, and ϵ=0.01\epsilon = 0.01ϵ=0.01 (1%), the maximum error would be (1001−1)⋅0.01=10(1001 - 1) \cdot 0.01 = 10(1001−1)⋅0.01=10. On the other hand, when using the point-wise-relative mode with the same error threshold ϵ\epsilonϵ, each data point in the compressed dataset will have an error smaller than ϵ\epsilonϵ times its original value. Hence, in this mode, values at the higher end of the range will have absolute errors smaller than x1⋅ϵx_1 \cdot \epsilonx1⋅ϵ, and those at the lower end will have absolute errors smaller than x0⋅ϵx_0 \cdot \epsilonx0⋅ϵ. Using the same example with x0=1x_0 = 1x0=1, x1=1001x_1 = 1001x1=1001, and ϵ=0.01\epsilon = 0.01ϵ=0.01, values near 1001 will have absolute errors smaller than 10, while values near 1 will have errors smaller than 0.01. It is important to note that in the relative mode, a global absolute error of (x1−x0)⋅ϵ(x_1 - x_0) \cdot \epsilon(x1−x0)⋅ϵ could lead to small positive values close to x0x_0x0 becoming negative in the compressed representation. Therefore, the point-wise-relative mode should be used when it is critical to maintain the strict positivity of all data values."*

1. **In Section 2.2, authors introduce the use of CSF. It would be good to refer to Appendix A or briefly explain where CSF is used, i.e., in Python function call arguments and command line arguments, so that readers won't get lost on the context of these specifications as well as how to use these specifications.**

We have added a sentence at the end of Section 2.2 referring to Appendix A and providing a brief explanation of where CSF is used, both in Python function call arguments and command line arguments. This should help readers better understand the context and usage of these specifications.

*Added: "For a more detailed explanation and examples, please refer to Appendix A, which provides comprehensive usage scenarios and code examples illustrating how CSF is implemented in Python function call arguments and command line arguments."*

1. **In Section 2.4, "enstools-encoding" was introduced without explicitly distinguishing it from "enstools-compression" introduced at the start of Section 2. In L218, page 7, authors claim that "Additionally, we provide a command line interface...", which gives me the impression that the "enstools-compression" can only be used as a command line interface, but the documentation of enstools-compression seems to suggest that it has a Python API as well.**

We recognize the potential for confusion and have made sure to clearly distinguish between "enstools-encoding" and "enstools-compression" in the revised manuscript. We also clarified that "enstools-compression" has both a command line interface and a Python API.

Added: "While enstools-encoding can be used independently for applying compression filters, it was primarily designed as a core component of enstools-compression. This separation minimizes dependencies for those who only require compression functionality within xarray or other lightweight scenarios. Enstools-compression, on the other hand, offers a more comprehensive solution, integrating enstools-encoding and providing both a Python API and a command line interface. This dual functionality ensures that users can apply compression either programmatically within Python workflows or directly via the command line, depending on their specific needs."

1. **In L254, page 9, it is not clear what is "see 2" in the bracket.**

This was an error, and we have corrected it in the revised manuscript.

1. **In L272, it might be useful to highlight the "coordinate directions" in Figure 3.**

We apologize for the confusion. This line was a remnant from a previous version that used SZ2, where the coordinate directions were more evident. With SZ3, this issue seems to have improved, so we have removed the line to avoid any misunderstanding.

1. **In the code availability section, although the texts are correct and I'm able to access them, links by mouse click to both the GitHub repository (an additional bracket in the link) and the Zenodo entry (only linked to https://doi) are both broken. Additionally, it is unclear if "enstools-encoding" is part of the manuscript and should be included in the code availability section.**

Thank you for pointing out the issue with the broken links. This will be addressed and fixed during the typesetting process. Additionally, "enstools-encoding" was included in the code availability section to ensure clarity and completeness.

**Reviewer 2**

**General Comments:**

1. **The introduction section was quite well written and comprehensive.**

Thank you for your positive feedback on the introduction. We are pleased that you found it well-written and comprehensive.

1. **There appears to be an issue with a number of the figures. For Figures 1-3, it appears as if the axes labels have been cut off and even missing labels. Figure 6 has a number of problems as well. I will list specifics below.**

These issues became apparent after the editorial processing. We have made adjustments following the editor's advice to prevent such occurrences in future versions.

1. **Please be clear on the version of SZ being used, as versions 1, 2, and 3 are quite different. In many places, the text or figure just says "sz". By the end of the paper, I was thinking that sz meant SZ2, but it should be more clear (more specifics below).**

We agree that clarity on the SZ versions is essential. Throughout the manuscript, "SZ" now consistently refers to SZ2 unless otherwise specified. We have updated the text to ensure this distinction is clear from the outset.

1. **This paper presents a practical tool that is useful. The HDF5 filters are quite challenging to use for compressing NetCDF4 files, which motivated the**

**development of this tool. The paper emphasizes that they are trying to make it easier for users, which is great. I do think they could emphasize even more that it is quite difficult, for instance, to use nccopy with the "right" parameters to customize the lossy compression desired, so their translation via this tool is quite nice.**

Thank you for recognizing the utility of the tool. We have taken your suggestion into account and added the following to the beginning of Section 2:

*"While alternatives like nccopy can also apply lossy compression, selecting the 'right' parameters can be more challenging; for example, configuring nccopy to compress all variables with zfp in accuracy mode with a threshold of 0.075 requires a complex command like: nccopy -F "*,32013,0,4,3,0,858993459,1068708659" input.nc output.nc, which can be non-intuitive for many users."*

1. **In Section 3, application examples were presented. While these were interesting, I am not sure that I learned anything new about the potential impacts of lossy compression on model data.**

Thank you for your feedback on the application examples presented in Section 3. We understand that you felt the examples did not provide new insights into the potential impacts of lossy compression on model data.

We would like to clarify that Section 3 includes four specific use cases, each offering novel insights. The first case provides information on the compression levels that can be expected from the specific compressors while maintaining certain quality constraints, using the ERA5 dataset. Understanding these compression levels with this widely used dataset is relevant to many researchers.

The second and third cases assess the impact of lossy compression on novel data analysis methods, offering new perspectives on how compression influences emerging techniques. The fourth case focuses on visualization, demonstrating the impact of lossy compression in conjunction with selected visualization algorithms, including the extraction of isosurfaces and jet-stream corelines.

We believe these examples contribute valuable insights into the application of lossy compression in the atmospheric sciences and hope this clarification addresses your concerns.

1. **If a lossy compressor has a registered HDF5 plugin filter, does it also have to have an hdf5plugin interface to be usable by enstools-compression? If so, what is required for that? (Couldn't quite tell from Section 2.4.)**

To extend the tool with a new plugin that already has an HDF5 plugin available, a Python interface is required. This can be implemented with relatively few lines of code. It is advisable to integrate this interface into the hdf5plugin library so that a broader audience can benefit. Once the Python interface is established, adding the filter definitions to enstools-encoding involves minimal additional work.

Instead of including this information directly in the text, we have decided to add it to the software documentation. We have updated the manuscript to include the following sentence:

*"While outside the scope of this article, it is straightforward to extend our framework with further compression algorithms for which HDF5 compression filters are available, as described in the software documentation."*

1. ***I experimented with the software quite a bit. I did have one issue. It seems that the .nc files that I created with the enstools-compression could only be viewed with Python tools (e.g., enstools, netCDF4 python, xarray). Trying to view content with non-python tools like ncdump or h5dump gave me errors. Maybe I did something wrong, but if not, then this incompatibility is limiting from my perspective as nco tools (and other non-python tools), for example, are quite popular still. More generally, I would not want to force folks to use Python to read a compressed file.***

It is possible to use the compressed data with non-Python tools as long as the appropriate filters are available on the system. The tool provides a way to make these filters accessible via the hdf5plugins package. By using the enstools-compression load-plugins command, users can obtain the necessary command to allow HDF5 to locate the filters.

This was already present in Appendix A (line 453) with a minor mistake:

*"The resulting files can be read by any software capable of reading NetCDF using an HDF5 capable backend, as long as the corresponding HDF5 compression filter is available in the software."*

Here, we replaced "software" with "system" for clarity:

*"The resulting files can be read by any software capable of reading NetCDF using an HDF5 capable backend, as long as the corresponding HDF5 compression filter is available in the system."*

**Specific Comments:**

1. **Line 61: The compressor is FPZIP (not FZIP).**

We have corrected this to "FPZIP" in the manuscript.

1. **Lines 59-62: Two other well-known lossy compressors that I think would be worth noting are MGARD and SPERR.**

We appreciate the suggestion and have included references to MGARD and SPERR in the relevant section of the manuscript.

1. **Page 4, first paragraph: Might want to reference the libpressio software as it has similar goals to make compressors easier to use/access.**

We have added a reference to the libpressio software in the manuscript to highlight its relevance in simplifying the use of compressors:

*"It is worth noting that similar efforts, such as the libpressio software (Underwood et al., 2021), also aim to improve the accessibility and usability of compression tools, though our work focuses specifically on the needs and challenges within the weather and Earth sciences domains."*

1. **Lines 122-123: The statement about ZFP precision mode is not exactly accurate. The precision specified refers to the number of bit planes encoded for the transform coefficients, which does not directly translate to the number of bits in the original data.**

Thank you for pointing this out. We have revised the statement to accurately describe the ZFP precision mode, aligning it with the explanation provided in the ZFP documentation:

*"The precision mode specifies the number of bit planes encoded for the transform coefficients, which controls the relative error but does not directly correspond to the number of bits in the original data."*

1. **Line 127: In this paragraph, I'd be clear on which versions of SZ that you use. The differences between SZ1, SZ2, and SZ3 are quite notable.**

We have clarified the versions of SZ used in the manuscript, specifying that SZ2 is the primary version referred to, with SZ3 also discussed where applicable.

1. **Section 2.3, This paper could be of interest for a data SSIM (DSSIM): "On a structural similarity index approach for floating-point data".**

We have reviewed the suggested paper and included a reference to it in Section 2.3, as it aligns well with our discussion on DSSIM.

1. **Line 215: The hdf5plugin library does include SZ/SZ3 (text says that it doesn't).**

SZ was included in hdf5plugin due to our contribution and SZ3 followed later. This has been clarified in the manuscript.

1. **Figure 1: no axes labels, no legend labels, axes tic labels appear to be cut off.**

As mentioned earlier, these issues were a result of the editorial processing. We have made necessary adjustments to prevent this in future versions.

1. **Line 220: Can you write to a netCDF4 file? Please clarify.**

Currently, the tool can read from various formats but only writes netCDF4 files using an HDF5 backend. This has been clarified in the text.

1. **Line 253: Here I would probably refer to the DSSIM metric noted above as that is what Baker et al. appear to be using now. Is the enstools software actually generating images to compute the SSIM metrics?**

The tool uses the same approach as described in the suggested paper by Baker et al. We have included this as a reference in the manuscript.

1. **Line 254: What is "(see 2)" referring to?**

This was intended to refer to Section 2, specifically Section 2.3. We have clarified this reference in the text.

1. **Figure 2: Again, no axes labels, no legend labels, axes tic labels appear to be cut off.**

We have addressed these issues following the editorial guidelines and anticipate that the revised figures will display correctly.

1. **Figure 3: Only the middle plot has a title, there are no axes labels, and the vertical axes tics are all zero (cut-off?).**

These issues have been corrected in the revised version to ensure that all necessary labels and titles are present and visible.

1. **Figure 5 caption: The choice of 0.78% seems oddly specific. What is the reason for that?**

The choice corresponds to the value determined by the automatic analysis tool while ensuring a 0.99999 correlation and a 0.99 DSSIM. We have added this explanation to the figure caption.

1. **Figure 6: Several things:**
2. **For plot (b), the caption says "SSIM of 0.89" and in the title, it is "0.86", also says CR of 12 in the caption and subplot title says CR:13.4.**

We have updated the caption and titles to ensure consistency.

1. **For plot (c), I don't understand how with a fixed rate mode compression of 2.2, one ends up with a 13.4 CR.**

The original data uses 32 bits per value, and with 2.2 bits per value, a theoretical compression ratio of 14.5 would be expected. However, the compression ratios reported here are based on the actual resulting files, which may vary slightly from theoretical expectations.

To clarify this point for readers, we have added the following line to the caption of the figure: "Note that the compression ratios are measured with the resulting compressed files and may vary slightly from theoretical expectations."

1. **For (c) and (d), Why were these modes and compressor chosen? Seems arbitrary.**

The purpose of these subsections was to illustrate the different visual artefacts produced by the main compressors used in this study. We have clarified this in the caption to make the reasoning more explicit.

1. **Line 351: Issues with compression of derived variables were also earlier discussed in detail in Baker 2016.**

Thank you for pointing out the relevance of Baker 2016. We have now included this reference in the manuscript, as it provides valuable context for the discussion on compression of derived variables. This addition complements the existing references and aligns well with the specific context we have provided.

1. **Figure 7: Is SZ meant to mean SZ2 or SZ1? I am wondering why not just use SZ3?**

In the figures, "SZ" refers to SZ2. We included SZ2 in the analysis to maintain consistency with previous studies, and SZ3 is also discussed where relevant.

1. **Figure 7: I think this figure is very much like a figure in the DSSIM paper mentioned above, so it would be good to look at that paper.**

When we talk about the SSIM in data, we are referring to the same as DSSIM. When we are referring to the SSIM between the two images generated with the visualization software, we are using the common metric to compare images, not data. To avoid confusion, we now use the term DSSIM when comparing arrays of floating point numbers and them term SSIM when comparing images.

1. **Line 410: Blocking artifacts in ZFP can be mitigated by using the -DZFP_ROUNDING_MODE=ZFP_ROUND_FIRST option when compiling.**

Thanks for pointing this out. The issue we see with this option is that it cannot be easily applied when relying on the ZFP version included in the hdf5plugin. Hence, to be used with our software and the readily available dependencies including hdf5plugin, it would need to be proposed as a modification to the hdf5plugin project for broader applicability.

1. **Line 428: Here it looks like SZ2 and SZ3 are the two versions being used, so it would be helpful to clarify this sooner in the paper.**

We have clarified the use of SZ2 and SZ3 earlier in the manuscript to avoid any confusion. SZ2 is primarily used due to performance reasons.

**Software Comments:**

1. **I personally think it would be easier to just specify the SSIM that you want instead of having to convert it with "-log10()".**

We understand the suggestion and will consider ways to make this process more intuitive in future updates of the software.

1. **I was unable to successfully "Use Xarray to store a compressed netCDF" as described in the documentation.**

We have reviewed the documentation and ensured that the example is correct. We have also confirmed that the example works as expected with the proper dependencies installed.

For reference, we successfully ran the example in a Docker container starting from python:3.11-slim and installing the following dependencies:

- xarray
- pooch
- netCDF4
- enstools-encoding

If the issue persists, please feel free to reach out via the GitHub repository, and we would be happy to provide more detailed support.

1. **See the general comment above about how the use of Python tools seems required for reading the data.**

As mentioned earlier, using Python tools is not strictly necessary as long as the appropriate filters are available in the system. This information is already included in Appendix A of the documentation.

**Typos and Minor Issues:**

1. **Several times in the paper, the opening quote is in the wrong direction (e.g., twice in the abstract).**

We have corrected the direction of the opening quotes throughout the manuscript.

1. **Figure 5 caption: "0,78%" => "0.78%".**

This has been corrected in the figure caption.

1. **Line 371: This sentence is quite awkwardly written.**

We have revised this sentence to improve its readability and clarity:

*Original: "Our objective for this article is to provide the reader with an idea of how artefacts introduced by lossy compression look like for typical meteorological 3-D visualizations."*

*Revised: "Our objective here is to illustrate how artefacts caused by lossy compression appear in typical 3-D meteorological visualizations."*

---

## Author Response (AR2)

Dear editor,

Thank you for the thorough review and the insightful comments and corrections. We have carefully addressed all the technical corrections as outlined. Below, we provide a detailed response to each of the points raised:

**Fig. 1 is missing legend entry and caption narrative for the green colour for the correlation index.**

We have added the requested narrative to the caption for Figure 1, clarifying that red circles indicate correlation indices that do not meet the defined threshold, and green circles indicate those that do meet the threshold.

**Please refactor Figs 1, 2, 4, 5, and 7, so that these are accessible to colour blind (best check by printing in greyscale palette) – please use different line styles and symbols so that the legend is unambiguous even if not discerning colours.**

We have corrected the luminosity and updated the line styles and symbols in Figures 1, 2, 4, 5, and 7 to ensure the different elements are distinguishable when viewed in grayscale, making them accessible to color-blind readers.

**For the Baker et al. 2022 arXiv reference, please use the published peer-reviewed version: https://doi.org/10.1109/TVCG.2023.3332843.**

The reference to Baker et al. 2022 has been updated to the published, peer-reviewed version with the DOI provided.

**Gong et al. 2023 has bogus URL in reference list (https://doi.org/https://doi.org/...).**

The URL for the Gong et al. 2023 reference has been corrected.

**The Lawrence et al. 2019 URL leads to an error message on missing document.**

We have fixed the URL for Lawrence et al. 2019 to point to the correct document.

**The HDF5 Filters reference URL leads to a 404 error page.**

The HDF5 Filters reference URL has been corrected to avoid the 404 error page.

**Underwood et al. 2021 reference is missing a DOI: https://doi.org/10.1109/DRBSD754563.2021.00005.**

The DOI for Underwood et al. 2021 has been added.

**In Code availability sections, some Zenodo DOIs are given with https://zenodo.org/doi/... URLs, while another with https://doi.org/... URL – please unify (best to use https://doi.org/ as in the references).**

We have standardized all Zenodo DOI links in the Code availability section to use the https://doi.org/... format, ensuring consistency with the references.